# Bridge Thinking and Acting: Unleashing Physical Potential of VLM with Generalizable Action Expert

## Abstract

Although Vision-Language Models (VLM) have demonstrated impressive planning and reasoning capabilities, translating these abilities into the physical world introduces significant challenges. Conventional Vision-Language-Action (VLA) models, which integrate reasoning and action into a monolithic architecture, generalize poorly because they are constrained by scarce, narrow-domain data. While recent dual-system approaches attempt to decouple "thinking" from "acting," they are often constrained by semantic ambiguities within the action module. This ambiguity makes large-scale, cross-task training infeasible. Consequently, these systems typically necessitate fine-tuning on newly collected data when deployed to novel environments, and the cooperation mechanism between the two systems remains ill-defined. To address these limitations, we introduce, for the first time, a framework centered around *a generalizable action expert*. Our approach utilizes sparse 3D trajectories as an intermediate representation, effectively bridging the high-level planning capabilities of the VLM with the low-level physical action module. During the planning phase, the VLM is only required to generate coarse 3D waypoints. These waypoints are then processed by our generalizable action expert, which refines them into dense, executable action sequences by sampling real-time point cloud observations of the environment. To promote training efficiency and robust generalization, we introduce a novel "Action Pre-training, Pointcloud Fine-tuning" paradigm. Our method combines the broad generalization capabilities of VLMs in visual understanding and planning with the fine-grained, action-level generalization of action expert. Through extensive experiments, we demonstrate that our method exhibits high-quality results and strong generalization across diverse visual domains, camera viewpoints, and natural language instructions, enabling zero-shot deployment without further fine-tuning.

## 1 Introduction

*"Learning is about generalization, not memorization."*

— Herbert A. Simon, *Models of Thought* (1979)

Vision-language models (VLMs) (Bai et al., 2025; Chen et al., 2024b; Wang et al., 2025b; Comanici et al., 2025) have demonstrated powerful capabilities in visual understanding, spatial reasoning, and task planning. However, translating these abilities to the physical world presents unprecedented challenges. A prevailing strategy to bridge this gap is the Vision-Language-Action (VLA) model (Kim et al., 2024; Brohan et al., 2022; Zitkovich et al., 2023), which integrate reasoning and action into a monolithic architecture. However, this approach introduces a critical paradox: the fine-tuning required for effective robotic control, typically on scarce, narrow-domain robotics data, often causes catastrophic forgetting on VLM. This process inadvertently erodes the VLM's foundational knowledge, leading to the poor generalization commonly observed in VLAs and undermining the primary motivation for the use of VLMs.

A key reason behind this phenomenon is that VLA models often conflate the high-level reasoning and low-level execution. While recent methods (Black et al.; Li et al., 2025; Huang et al., 2025;

Li et al., 2024) have moved towards decoupling "thinking" and "action" into distinct modules typically with dual-system frameworks, a persistent challenge still remains. The action module is still required to interpret semantically rich information, such as visual features (Bjorck et al., 2025) or semantic embeddings (Huang et al., 2025) from a VLM. As a result, the action policy is caught in a dilemma, as it faces the conflicting demands of being lightweight enough for real-time execution while also being responsible for interpreting complex high-level semantic information. This semantic burden on the execution policy renders large-scale, cross-task training infeasible and severely restricts its generalization capability. Consequently, these approaches did not fundamentally resolve the aforementioned key challenges in VLA models.

To address these limitations and achieve a true decoupling of planning and execution, we introduce a novel framework centered around a generalizable action expert. We architect the system so that this expert communicates with the VLM planner via an explicit representation: a sparse sequence of coarse 3D waypoints. This design choice places the planning task squarely in the VLM's comfort zone, as generating simple geometric coordinates instead of complex embeddings is well-aligned with its inherent capabilities. This approach minimizes the need for extensive fine-tuning, thereby preserving the VLM's rich world knowledge and maximizing its generalization.

For the action expert itself, this explicit guidance is transformative. It is liberated from the burden of complex semantic interpretation, shifting its role from a challenging, reasoning-dependent semantic-to-action mapping to a more tractable geometric refinement task. Guided by the coarse waypoints, the expert leverages real-time point cloud observations to refine the sparse trajectory into a dense, executable action sequence. To endow our expert with this capability, we introduce a novel "Action Pre-training, Pointcloud Fine-tuning" paradigm. This entire framework, analogous to human motor control, allows the action expert to focus solely on robust, real-time execution. Our method synergistically combines the perceptual and reasoning generalization of VLMs with the fine-grained, motion-level generalization of our generalizable action expert. As extensive experiments confirm, this synergy results in a system with powerful generalization, enabling robust zero-shot deployment without the need for further fine-tuning. In conclusion, our contributions are summarized as follows:

- We introduce a framework centered on a generalizable action expert that uses sparse 3D trajectories as a clean interface. This architecture fully decouples high-level VLM planning from low-level motor control. *To the best of our knowledge, this is the first attempt to train a generalizable expert that can be deployed without requiring any task-specific fine-tuning*.

- We propose the "Action Pre-training, Pointcloud Fine-tuning" strategy. This method enables our action expert to generalize by focusing it on geometric trajectory refinement rather than semantic interpretation.

- Our system demonstrates remarkable generalization across diverse experiments. We validate its practical viability through successful zero-shot deployment without any in-domain fine-tuning

## 2 RELATED WORKS

### 2.1 SPATIAL REASONING ABILITY IN VISION-LANGUAGE MODELS

Equipping vision-language models (VLMs) with spatial reasoning capabilities has emerged as a prominent research focus. Pre-trained on large-scale datasets, recent VLMs (Bai et al., 2025; Chen et al., 2024b; Comanici et al., 2025) have demonstrated considerable ability in understanding 3D spatial relationships, achieving competitive performance on spatial reasoning benchmarks (Zhang et al., 2021; Azuma et al., 2022; Ma et al., 2022).A number of studies (Wu et al., 2025a; Cai et al., 2024; Zhu et al., 2024; Chen et al., 2024a; Yuan et al., 2024; Song et al., 2025; Yang et al., 2025) have sought to equip large models with powerful spatial understanding and reasoning. Their approaches typically involve either fine-tuning on extensive 3D datasets or directly incorporating explicit 3D information. To prove the widespread validity of our framework, we used a widely adopted, general Vision-Language Model (VLM) as our baseline, rather than models specifically fine-tuned for 3D data. Our paradigm is exceptionally flexible and is compatible with almost all current VLM models.

### 2.2 VISION-LANGUAGE-ACTION MODELS

Recent advances in Vision-Language-Action (VLA) models have explored diverse architectures: Conventional VLA models (Brohan et al., 2022; Zitkovich et al., 2023; Cheang et al., 2024; Ha et al.,

2023; Kim et al., 2024; Black et al.; Wen et al., 2025b; Team et al., 2024) typically employ a single, end-to-end architecture to map vision and language directly into an action space. Recent dual-system frameworks employ a hierarchical architecture consisting of a high-level planning model and a low-level action expert. These two components communicate through various intermediate representations, such as trajectories (Li et al., 2024; de Bakker et al., 2025; Huang et al., 2025), visual features (Li et al., 2025; Bjorck et al., 2025), or attention mechanisms (Black et al.; Intelligence et al., 2025; AgiBot-World-Contributors et al., 2025). Although this design offers a degree of decoupling compared to monolithic VLA architectures, the semantically ambiguous nature of these intermediate representations poses a significant challenge for training the downstream action expert. The expert becomes burdened with interpreting a portion of the semantic and task-specific information, which impedes large-scale training. Our approach effectively alleviates this issue by using explicit, sparse 3D trajectories as a clear and unambiguous intermediate representation.

## 2.3 GENERALIZABLE ACTION EXPERT

Developing a generalizable action expert is a longstanding challenge in robotics. Commonly used action experts (Chi et al., 2023; Zhao et al., 2023) are computationally efficient but have limited model capacity, causing them to overfit and perform poorly in multi-task scenarios. To reduce the heavy semantic planning burden on the action expert and improve its generalization, a popular approach is to supply it with guidance signals. This guidance, which can take the form of object poses (Deng et al., 2020), keypoints (Manuelli et al., 2019), affordance Wu et al. (2025b), and semantically segmented point clouds (Zhu et al., 2023), acts as an approximate action outline, simplifying the problem and allowing the policy to concentrate on low-level refinement instead of high-level strategic planning. Although these approaches improve generalization to some extent, they still require fine-tuning for specific scenes and fail to adapt to truly novel environments. In our method, we use sparse 3D end-effector pose trajectories as clear guidance, which are then refined into executable and accurate action sequences by processing point clouds captured from the environment in real time.

## 3 METHOD

Figure 1 shows the full pipeline of our method. We utilize 3D spatial trajectories to serve as the bridge between our high-level VLM and low-level action expert. The process begins with the VLM, which reasons about 2D keypoints and leverages depth data to generate two key outputs: a sparse set of 3D waypoints and the final end-effector pose at the target keypoint. After transforming these from the camera frame to the robot's base frame, we use a B-spline to interpolate them. This step converts the sparse points into a continuous and smooth end-effector pose trajectory, from which guidance signals are sampled for the Action Expert, as we will detail in section 3.1 and 3.2.

Traditional Vision-Language-Action (VLA) models typically predict coordinates in the robot's base frame. We argue this approach is inherently flawed, as it implicitly forces the Vision Language Model (VLM) to learn a complex camera-to-robot transformation, often without the necessary camera priors. Such a task runs counter to the VLM's vision-centric nature, encouraging shortcuts like memorizing spatial coordinates rather than truly understanding spatial relationships. This tendency explains the poor generalization commonly seen in these models. Our approach overcomes this by predicting waypoints directly in the camera frame. This formulation is far more intuitive, as it aligns directly with the VLM's image-based pre-training and enables the model to fully preserve and leverage its powerful prior knowledge.

## 3.1 LIFT VLM'S REASONING POWER TO 3D SPACE

Recent work has demonstrated that visual-language models pre-trained on internet-scale data exhibit strong abilities in 2D object localization and spatial reasoning. However, enabling VLMs to reason about the target robot poses in a zero-shot manner remains challenging. To address this, we fine-tune a VLM with a small amount of carefully annotated data. Specifically, we identify and select keyframes for annotation at moments where the gripper's kinematic state changes, such as during opening or closing, as these signify critical points of interaction. This annotation pipeline, described in Figure 2, allows the model to estimate 3D spatial poses while preserving its original linguistic and reasoning capabilities.

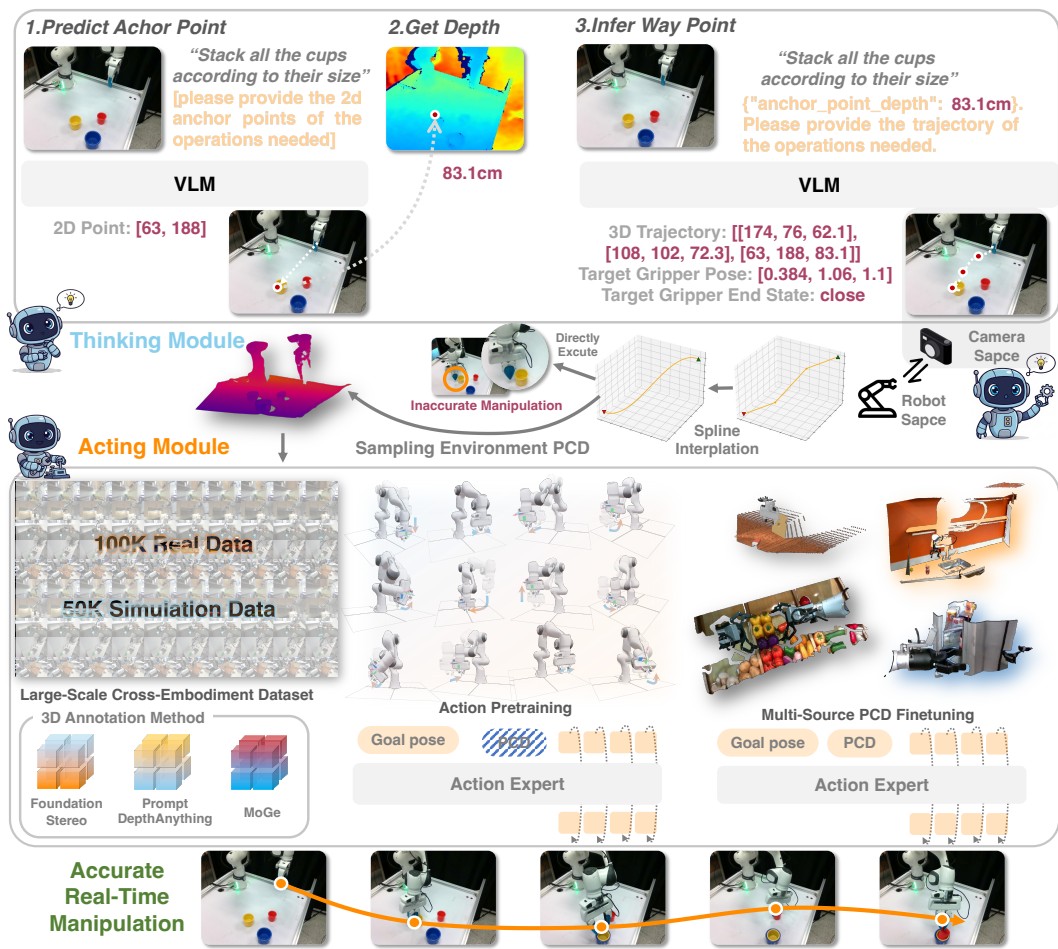

Figure 1: **The pipeline of our proposed method.** Our approach begins with a VLM predicting a sparse set of 3D waypoints directly in the camera frame, preserving its vision-centric knowledge. These sparse points are then transformed and interpolated via a B-spline into a continuous and smooth pose trajectory to provide dense guidance for a low-level action expert.

We annotate data based on gripper states, selecting keyframes where the state changes to construct a supervised fine-tuning (SFT) dataset. The process is as follows: First, the VLM predicts the 2D anchor point coordinates for grasping or placing a target object. Using depth information, we obtain the corresponding 3D coordinates $(u, v, d)$ of the target. This information is then fed into the VLM to infer waypoints along the motion path, along with a target end-effector pose. Finally, we apply spline interpolation to these waypoints and the target pose, generating a continuous end-effector pose trajectory that provides guidance for the subsequent action expert.

## 3.2 TRAINING GENERALIZABLE ACTION EXPERT

Developing our generalizable action expert required overcoming two key challenges: **the scarcity of high-quality point cloud data**, and **the need for a training paradigm that maximizes efficiency without sacrificing generalization**. Our approaches to these fundamental problems are detailed in Section 3.2.1 and 3.2.2.

### 3.2.1 DATA PREPARATION

Training a well-generalized action expert model requires a large volume of high-quality 3D point cloud annotations from both real-world and simulated environments. In simulation, obtaining high-fidelity point clouds is straightforward due to precise depth and camera data. We replayed

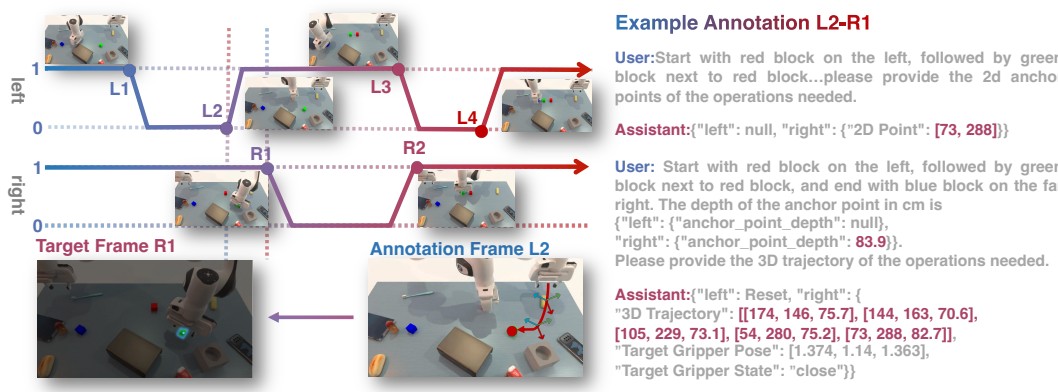

Figure 2: **Overview of our data annotation pipeline.** We construct our SFT dataset by first selecting keyframes based on gripper state changes.

trajectories from multiple simulators, including RoboTwin (Mu et al., 2025; Chen et al., 2025), CALVIN (Mees et al., 2022), LIBERO (Liu et al., 2023) and RLBench (James et al., 2019), yielding a total of 50k trajectories accompanied by high-accuracy point clouds and joint poses.

In contrast, real-world datasets present significant challenges. Most existing robot manipulation datasets lack high-quality depth annotations and accurate camera calibration parameters. Their depth information typically comes from depth sensors, resulting in sparse and incomplete depth maps. This severely limits their utility for training general-purpose robot learning models.

To overcome this limitation, we have re-annotated several real-world robot datasets with more accurate and dense depth information. Specifically, for the DROID dataset (Khazatsky et al., 2025), which provides stereo imagery, we employ FoundationStereo (Wen et al., 2025a) to generate high-quality stereo depth estimates, significantly enhancing the accuracy and density of the depth annotations. For AGIBOT (AgiBot-World-Contributors et al., 2025), the raw depth maps from its native camera are often sparse and of low quality. To address this limitation, we employ Prompt-DepthAnything (Lin et al., 2025) to perform depth completion, generating dense and high-quality depth information for downstream tasks.

To account for real-world scenarios where high-precision point cloud data may be unavailable, we further utilized MoGe (Wang et al., 2024; 2025a) to re-annotate point clouds for a subset of the BridgeV2 (Walke et al., 2023) and pre-mentioned simulation data. This step allows us to simulate conditions lacking reliable depth information and test our model's robustness.

To improve point cloud downsampling efficiency, we adopt the cropping strategy from 3D Diffusion Policy (Ze et al., 2024). For simulated data, we directly use ground-truth segmentation IDs. For real-world data, we generate foreground masks through a custom pipeline: initial masks from RoboEngine (Yuan et al., 2025) are refined using temporal consistency tracking and integrated with Segment Anything Model 2 (SAM 2) (Ravi et al., 2024) to ensure accuracy and temporal consistency.

### 3.2.2 ACTION PRETRAINING, POINTCLOUD FINETUNING

Training our generalized action expert with extensive point cloud and trajectory data concurrently poses a significant efficiency challenge. We identified that the expert's role can be decomposed into two core skills: basic trajectory-following and environment-aware trajectory refinement using point clouds. To avoid the high cost and suboptimal learning that can result from training these coupled skills together, we introduce the "Action Pre-training, Point Cloud Fine-tuning" paradigm. We first pre-train the expert on large batches of pure trajectory data (up to a batch size of 31,824) to master motion following, and then fine-tune it with point cloud data to learn refinement. Our experiments confirm this decoupled approach achieves faster convergence and substantially improves data utilization efficiency. Now we detail the architecture of our proposed model and its training process:

**Model Architecture** Our action expert's architecture is inspired by the design of 3D Diffusion Policy (Ze et al., 2024). It integrates multimodal sensory inputs to guide the action generation process. The full set of conditioning inputs $\mathcal{C}_A$ for the action expert at any given timestep $t$ is defined as:

$$\mathcal{C}_A = \{S_t, \mathcal{P}_g, O_{pcd}\} \tag{1}$$

where $S_t$ represents the robot's proprioceptive state, including information like joint positions and gripper state. $\mathcal{P}_g$ is the guidance pose, which is sampled from the continuous trajectory $\mathcal{T}(t)$ generated by the high-level VLM. $O_{pcd}$ is the cropped point cloud observation of the local environment.

Each component of $\mathcal{C}_A$ is processed by a dedicated encoder (MLPs for $S_t$ and $\mathcal{P}_g$, and a PointNet-based encoder for $O_{pcd}$) to produce a final conditioning feature vector $f_t^A$.

$$f_t^A = \text{concat}(f_t^s, f_t^g, f_t^{pc}) \tag{2}$$

**Conditional Diffusion Model Training** We model the action expert as a conditional diffusion policy. Instead of directly predicting an action, the policy learns to reverse a Gaussian diffusion process, iteratively refining a noisy action into a clean one, conditioned on $f_t^A$. The policy is parameterized as a noise prediction network $\epsilon_\theta$.

During training, we sample a ground-truth action $a_t^0$ from the expert demonstration dataset $\mathcal{D}$. We then create a noisy action $a_t^k$ by adding $k$ steps of Gaussian noise according to the noise schedule $\bar{\alpha}_k$:

$$a_t^k = \sqrt{\bar{\alpha}_k}a_t^0 + \sqrt{1 - \bar{\alpha}_k}\epsilon \tag{3}$$

where $\epsilon \sim \mathcal{N}(0, \mathbf{I})$ is random Gaussian noise. The noise prediction network $\epsilon_\theta$ is trained to predict the added noise $\epsilon$ based on the noisy action $a_t^k$, the diffusion step $k$, and the conditioning feature $f_t^A$. The learning objective is to minimize the L2 error on the predicted noise:

$$\mathcal{L}_{AE} = \mathbb{E}_{k\sim[1,K],a_t^0\sim\mathcal{D},\epsilon\sim\mathcal{N}(0,\mathbf{I})} \left[\|\epsilon - \epsilon_\theta(a_t^k, k, f_t^A)\|^2\right] \tag{4}$$

At inference time, an action is generated by starting with a random noise vector $a_t^K \sim \mathcal{N}(0, \mathbf{I})$ and iteratively applying the learned network $\epsilon_\theta$ to denoise it over $K$ steps, finally yielding a clean action $a_t^0$.

This formulation fits our "Action Pre-training, Point Cloud Fine-tuning" paradigm. During pre-training, the point cloud feature within $f_t^A$ is masked out, training the diffusion model to follow trajectories. During fine-tuning, the full conditioning vector is used, enabling the model to refine its actions based on environmental context.

## 4 EXPERIMENTS

### 4.1 EXPERIMENT SETUP

Our experimental validation spans both simulation, using the RoboTwin (Chen et al., 2025) and ManiSkill (Mu et al., 2021) benchmarks, and real-world hardware. In all experiments, our action expert is deployed in a zero-shot manner, while the VLM undergoes only a few steps of Supervised Fine-Tuning (SFT) to learn formatted output prediction and pose inference. We analyze the effect of the number of SFT steps on performance in Section 4.2, evaluating the entire spectrum from a zero-shot setup to extensive fine-tuning.

### 4.2 MAIN RESULTS

We present a comprehensive comparison of our model against existing generalist and expert models across 11 tasks in the RoboTwin benchmark, categorized into short, middle, and long horizons (Table 1). Our model surpasses popular generalist models on all tasks. For short and middle-horizon tasks, we achieve performance on par with the DP3 single-task expert model. Our primary advantage is demonstrated in long-horizon tasks that require VLM-based planning. On these tasks, where specialized expert models almost universally fail, our model shows exceptional capability, achieving a 60% average success rate. Notably, while other generalist models like Pi0 and RDT

require task-specific fine-tuning after multi-task training, our method does not. The fine-tuning steps for our VLM are detailed in Table 2.

Table 1: Performance comparison between our multi-task generalist model and various single-task expert models across short, middle, and long-horizon tasks. The best performance in each row is **bolded**, and the second-best is underlined.

| Category | Task Name | Generalist Model | | | Expert Model | | |
|---|---|---|---|---|---|---|---|
| | | **Ours** | **Pi0\*** | **RDT\*** | **ACT** | **DP** | **DP3** |
| **Short Horizon** | Click Bell | **0.93** | 0.44 | 0.80 | 0.58 | 0.54 | 0.90 |
| | Grab Roller | 0.97 | 0.96 | 0.74 | 0.94 | **0.98** | **0.98** |
| | Lift Pot | 0.95 | 0.72 | 0.84 | 0.88 | 0.39 | **0.97** |
| | Place Phone Stand | 0.39 | 0.35 | 0.15 | 0.02 | 0.13 | **0.44** |
| | **Avg.** | 0.81 | 0.62 | 0.63 | 0.61 | 0.51 | **0.82** |
| **Middle Horizon** | Handover Mic | 0.99 | 0.98 | 0.90 | 0.85 | 0.53 | **1.00** |
| | Place A2B Left | 0.38 | 0.31 | 0.03 | 0.01 | 0.02 | **0.46** |
| | Place Bread Basket | **0.65** | 0.17 | 0.10 | 0.06 | 0.14 | 0.26 |
| | Stack Block Two | **0.88** | 0.42 | 0.21 | 0.25 | 0.07 | 0.24 |
| | **Avg.** | **0.73** | 0.47 | 0.31 | 0.29 | 0.19 | 0.49 |
| **Long Horizon** | Blocks Ranking RGB | **0.78** | 0.19 | 0.03 | 0.01 | 0.00 | 0.03 |
| | Blocks Ranking Size | **0.53** | 0.07 | 0.00 | 0.00 | 0.01 | 0.02 |
| | Stack Block Three | **0.49** | 0.17 | 0.02 | 0.00 | 0.00 | 0.01 |
| | **Avg.** | **0.60** | 0.14 | 0.02 | 0.003 | 0.003 | 0.02 |

Table 2: Training Steps for Different Methods(GPU num*batchsize*steps).

| Method | Ours | ACT | DP | DP3 | Pi0 | RDT |
|---|---|---|---|---|---|---|
| **Training Steps (All tasks training)** | 8*32*1000 | 0 | 0 | 0 | 8*32*1000 | 8*32*1000 |
| **Training Steps (Single task fine-tuning)** | 0 | 32*10000 | 32*10000 | 32*30000 | 8*32*5000 | 8*32*5000 |

## 4.3 GENERALIZATION ABILITY

A critical limitation of conventional Vision-Language-Action (VLA) and diffusion-based models is their dependency on training data, which often leads them to "memorize" specific trajectories rather than learning truly generalizable skills. This deficiency becomes particularly pronounced in their failure to generalize to novel camera viewpoints, often requiring task-specific fine-tuning. To rigorously demonstrate our model's superior generalization, we conduct a multi-faceted evaluation. As detailed in Table 3, after fine-tuning on only 200 trajectories with varied camera angles, our model shows minimal performance drop on out-of-domain (unseen) perspectives, confirming its viewpoint invariance. Furthermore, Table 4 showcases its strong zero-shot generalization on tasks with novel colors, objects, and semantics, significantly outperforming the pi0 baseline using the same pre-trained model. To validate the cross-environment transferability of our action expert, we test its performance on the ManiSkill benchmark (Table 5), an environment whose data was entirely excluded from pre-training, thereby confirming its robust generalization.

Table 3: Camera View

| Task Name | Camera View | |
|---|---|---|
| | In-Domain | Out-of-Domain |
| Place A2B Left | 0.28 | 0.26 |
| Place A2B Right | 0.24 | 0.19 |
| Stack Block Three | 0.36 | 0.30 |
| Stack Bowl Three | 0.54 | 0.56 |
| Blocks Ranking Size | 0.21 | 0.14 |
| Blocks Ranking RGB | 0.52 | 0.48 |

Table 4: Zero-shot Results

| Category | Task Name | Ours | Pi0 |
|---|---|---|---|
| Color | Stack Block Two | **0.86** | 0.12 |
| | Blocks Ranking RGB | **0.69** | 0.32 |
| Object | Stack NumberBlocks | **0.38** | 0.00 |
| | Click Alarm Clock | **0.58** | 0.20 |
| Semantic | Place A2B Right | **0.39** | 0.23 |
| | Place A2B Randomly | **0.28** | 0.16 |

Table 5: Maniskill

| Task | Ours | DP3 | ACT |
|---|---|---|---|
| Push Cube | **0.89** | 0.83 | 0.81 |
| Stack Cube | **0.84** | 0.76 | 0.69 |
| Pull Cube Tool | **0.62** | 0.48 | 0.40 |

## 4.4 REAL WORLD EXPERIMENTS

Our robotic setup includes a single Franka Research 3 arm equipped with a UMI gripper (Chi et al., 2024). A third-person-view RealSense D435 camera is mounted in a fixed position to capture environmental observations at a resolution of 640×480 pixels. As shown in Figure 3, for real-world

evaluation, we designed six tasks that mirror our simulation experiments, categorized as short, middle, and long-horizon (two tasks per category). We collected 50 human demonstrations for each task, creating a multi-task dataset of 300 trajectories to fine-tune our model. During this process, only the Vision-Language Model (VLM) was updated, while the action expert remained frozen. In contrast, all baseline models were fine-tuned on a single-task basis. To ablate the role of our action expert, we also evaluate a VLM+IK baseline, which applies spline interpolation directly to the 3D waypoints predicted by the VLM for execution. For evaluation, each task was attempted 20 times to report the final success rate, as shown in Table 6. Further details of our real-world robot setting and task definitions can be found in Appendix A.3.

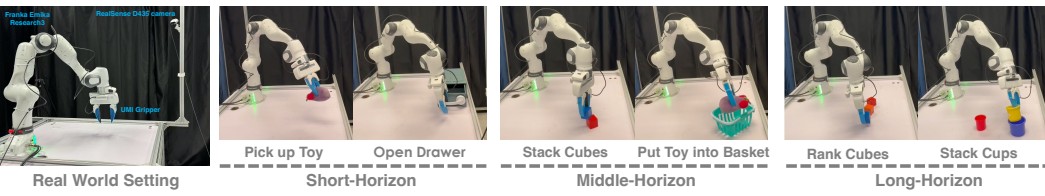

Figure 3: Real World Task Setting.

Table 6: Performance comparison of different methods across short, middle, and long-horizon tasks.

| Method | Short Horizon | | Middle Horizon | | Long Horizon | | Average |
|---|---|---|---|---|---|---|---|
| | Pick up Toy | Open Drawer | Stack Cubes | Put toy into Basket | Rank Cubes | Stack Cups | |
| ACT | 0.65 | 0.55 | 0.40 | 0.50 | 0.10 | 0.00 | 0.367 |
| DP | 0.85 | 0.75 | 0.40 | 0.45 | 0.15 | 0.00 | 0.433 |
| DP3 | 0.90 | **1.00** | 0.60 | 0.75 | 0.20 | 0.15 | 0.600 |
| OpenVLA | 0.85 | 0.40 | 0.35 | 0.55 | 0.45 | 0.20 | 0.467 |
| VLM+IK | 0.75 | 0.60 | 0.40 | 0.50 | 0.50 | 0.30 | 0.508 |
| VLM+DP(Origin) | 0.90 | 0.80 | **0.75** | **0.80** | 0.70 | **0.55** | 0.750 |
| VLM+DP(MoGe) | 0.85 | 0.75 | 0.70 | 0.60 | 0.60 | 0.30 | 0.633 |
| VLM+DP(PromptDepth) | **0.95** | 0.85 | **0.75** | 0.75 | **0.85** | 0.55 | **0.783** |

## 4.5 ABLATION STUDIES

### 4.5.1 ABLATION ON TRAINING STEPS

We investigated our model's performance across a spectrum of Vision-Language Model (VLM) fine-tuning, from zero fine-tuning to excessive fine-tuning that resulted in a significant degradation of language abilities in Table 7. We measured the success rate on a subset of our RoboTwin tasks with two approaches: 1) directly using Inverse Kinematics (IK) to execute trajectories generated by the VLM, and 2) utilizing our generalizable action expert to process the VLM's output. Concurrently, we benchmarked the VLM's language capabilities (e.g., using MMLU). The results clearly indicate that employing our generalizable action expert allows the model's performance to saturate much more rapidly, as clearly demostrated in Figure 4. This significantly reduces the required number of Supervised Fine-Tuning (SFT) steps for the VLM, thereby preserving its crucial language-based generalization capabilities.

Table 7: SFT step ablation.

| SFT Steps | 0 | 500 | 1000 | 1500 | 2000 | 2500 | 3000 | 3500 | 4000 |
|---|---|---|---|---|---|---|---|---|---|
| VLM + IK Avg | 0.04 | 0.26 | 0.32 | 0.34 | 0.40 | 0.43 | 0.47 | 0.52 | 0.52 |
| VLM + Expert Avg | 0.10 | 0.44 | 0.56 | 0.56 | 0.57 | 0.57 | 0.58 | 0.58 | 0.58 |
| Language Ability (MMLU) | 70.08 | 61.32 | 49.25 | 41.81 | 37.33 | 29.98 | 29.89 | 29.85 | 29.43 |

Table 8: Noise scale ablation

| Noise scale | Short Horizon | Middle Horizon | Long Horizon |
|---|---|---|---|
| 0.00 | 0.710 | 0.670 | 0.43 |
| 0.05 | 0.740 | 0.710 | 0.46 |
| 0.10 | 0.810 | 0.725 | 0.60 |
| 0.20 | 0.800 | 0.723 | 0.63 |
| 0.50 | 0.680 | 0.680 | 0.41 |

### 4.5.2 ABLATION ON DIFFERENT NOISE SCALES

We evaluated our model's robustness in Table 8 by adding noise of different scales to the goal pose, achieving optimal performance on the tasks in Table 1 at a noise scale of 0.1. This noise is intentionally introduced during the training of our generalizable action expert to simulate the inherent variability of VLM-generated trajectories, thereby improving the expert's ability to generalize. A careful balance is crucial: insufficient noise (or none at all) causes the expert to overfit to idealized

data, while excessive noise corrupts the training signal with unrealistic goals, ultimately degrading performance.

### 4.5.3 Ablation On Different Training Strategy

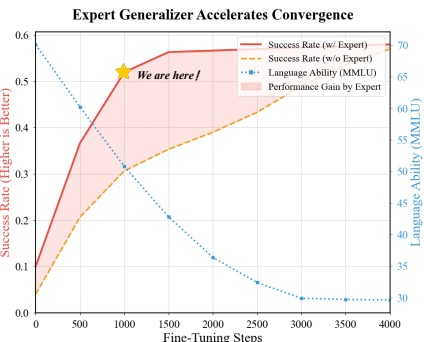

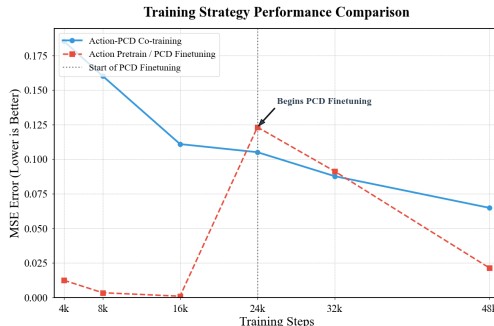

Figure 4: Ablation on training steps.

Figure 5: Ablation on training strategy.

To empirically validate our proposed "Action Pre-training, Point Cloud Fine-tuning" (APPF) paradigm, we compared its performance against the conventional end-to-end approach, where the model is trained jointly on trajectory and point cloud data from the start. The results unequivocally demonstrate the superiority of our decoupled strategy. During the action pre-training phase, by focusing solely on trajectory data, we circumvent the computational bottleneck of point cloud processing. This allows us to scale the training batch size to an unprecedented 32768, enabling the model to learn foundational motor skills at a dramatically accelerated rate and achieve faster convergence, as shown in Figure 5. Furthermore, our experiments reveal a clear and positive correlation between the volume of action data used in pre-training and the model's final success rate after fine-tuning. This substantiates our core hypothesis: building a robust motor foundation through large-scale action pre-training leads to a more data-efficient learning process and ultimately, a higher-performing generalized action expert.

### 4.5.4 Ablation On Different Pointcloud Source

As shown in Table 9, our ablation study on different point cloud sources reveals a critical trade-off. We used ground-truth simulation data (S), Foundation Stereo-annotated Droid data (F), PromptDepthAnything-annotated AGIBOT data (P), and MoGe-annotated BridgeV2/simulation data (M). We found that combining the P+M datasets, despite their relatively lower point cloud quality and a corresponding drop in simulation performance, significantly narrows the gap between simulation and real-world deployment.

Table 9: Ablation study of different model components across various task horizons.

|  | Short Horizon | Middle Horizon | Long Horizon | Real World Task |
|---|---|---|---|---|
| **F+P+M+S(main setting)** | 0.81 | 0.725 | 0.60 | 0.47 |
| **F+S** | 0.82 | 0.740 | 0.62 | 0.42 |
| **S** | 0.82 | 0.710 | 0.58 | 0.21 |

## 5 Conclusion

We have introduced a novel framework that resolves the critical trade-off between high-level reasoning and real-time control that has long constrained Vision-Language-Action models. By establishing a clean interface of sparse 3D waypoints, we achieve a true decoupling of the VLM's planning role from our action expert's execution role. This strategy, powered by our proposed "Action Pre-training, Pointcloud Fine-tuning" paradigm, liberates the action policy from semantic burdens and enables it to achieve unprecedented generalization. Our system's success in robust zero-shot deployment without any task-specific fine-tuning validates this approach, marking a significant step towards creating truly adaptable and scalable robotic agents.

ETHICS STATEMENT

We are committed to conducting research responsibly and in accordance with the ICLR Code of Ethics. Our study is based exclusively on publicly available datasets that are standard and widely adopted in the research community. The work presented does not involve human subjects, live animals, personally identifiable information, or any other form of sensitive data.

REPRODUCIBILITY STATEMENT

We are committed to ensuring the reproducibility of our results and have provided the following to support this goal:

- **Methodology:** Our model architecture and core methodology are described in detail in Section 3.
- **Implementation Details:** Comprehensive details regarding our datasets, robotic hardware, and training configurations are provided in Appendix A.2, Appendix A.3, and Appendix A.4, respectively.
- **Code Release:** Upon acceptance, we will release the complete source code and pre-trained models to the public to facilitate verification and future research.

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

## A APPENDIX

### USE OF LLMS

We used large language models (LLMs) only for minor assistance in polishing the language and adjusting the presentation of tables. No LLMs were involved in designing the methodology, conducting experiments, or analyzing results.

### A.1 DATA SOURCES AND COVERAGE

We integrate seven comprehensive data sources, LIBERO, RoboTwin 2.0, DROID, AgiBot World, CALVIN, RLBench and BridgeV2 into a unified framework, offering a diverse and robust foundation for learning manipulation skills across various scenarios and tasks.

**LIBERO**: A dataset specifically designed for lifelong robot learning and knowledge transfer across multiple tasks. LIBERO contains four task suites—LIBERO-Spatial, LIBERO-Object, LIBERO-Goal, and LIBERO-100—each targeting different aspects of generalization: spatial relationships, object variations, goal conditions, and long-term learning. The dataset provides multi-modal data, including RGB images from workspace and wrist cameras, proprioceptive states (joint positions, end-effector poses), natural language task descriptions, and PDDL-based scene annotations for high-level planning. With 130 tasks in total, LIBERO supports both pretraining (LIBERO-90) and evaluation (LIBERO-10) of lifelong learning algorithms, enabling research in compositional skill acquisition and cross-task knowledge transfer.

**RoboTwin 2.0**: A scalable data generator and benchmark for robust bimanual manipulation with strong domain randomization. It builds on the RoboTwin-OD object library, which includes 731 instances across 147 categories, and supports five dual-arm robotic platforms (Aloha-AgileX, ARX-X5, Franka, Piper, UR5), offering 50 collaborative tasks. The dataset also includes over 100K high-quality trajectories, providing a rich set of data with diverse conditions such as tabletop clutter, textures, lighting, and varied table heights.

**DROID**: A large-scale, real-world dataset containing **76,000 demonstration trajectories** (350 hours) across **564 scenes** and **86 tasks**, collected by **50 data collectors** in North America, Asia, and Europe. DROID emphasizes diverse, real-world manipulation tasks, offering multimodal data such as camera images, proprioceptive data, and language annotations. The dataset is particularly valuable for evaluating generalizable manipulation policies in dynamic, cluttered environments. Unlike more controlled datasets like **ManiSkill** and **RoboTwin 2.0**, DROID provides rich, real-world interactions, making it essential for training policies that must adapt to novel objects, environmental changes, and unexpected disturbances. Its robust data collection process ensures high-quality, real-world training data, significantly boosting policy performance and robustness.

**AgiBot World**: A large-scale, real-world manipulation platform that spans over 1M trajectories across 217 tasks, 87 skills, and 106 scenes, collected using a fleet of approximately 100 dual-arm humanoid robots (AgiBot G1). These robots are equipped with RGB-D cameras, fisheye cameras, visuo-tactile sensors, and optional 6-DoF dexterous hands. The dataset provides high-quality, multi-modal episodes with multi-view images, depth data, calibration, proprioception, and step-level language, emphasizing long-horizon, tool-use, deformable-object, and collaborative tasks in realistic environments, including domestic, retail, industrial, and office settings. The data collection process employs a standardized, human-in-the-loop pipeline (teleoperation, automatic validity checks, manual review, failure-recovery annotations) to ensure high-quality, multimodal data.

**CALVIN**: An open-source benchmark designed to evaluate long-horizon, language-conditioned robot manipulation. It is specifically built to test an agent's ability to solve complex, sequential tasks in a stateful environment where the world state persists between sub-tasks. CALVIN provides both a simulated environment and a real-world setup with a Franka Emika Panda arm, making it a robust platform for sim-to-real transfer research. The dataset includes multi-view RGB images, proprioceptive robot states, and corresponding natural language instructions for long-horizon task chains. Unlike episodic benchmarks that reset after each task, CALVIN's emphasis on compositional, stateful problem-solving makes it invaluable for developing and testing policies that require memory, sequential reasoning, and a deep understanding of language grounding.

**RLBench**: A large-scale and challenging benchmark designed to facilitate research in robot learning, offering a unified environment for both reinforcement learning (RL) and imitation learning (IL). Built on the CoppeliaSim physics simulator, it features over 100 unique, programmatically-defined tasks that cover a wide spectrum of manipulation challenges, from simple object placement to complex, multi-stage behaviors like stacking and opening containers. RLBench is not a static dataset but a dynamic data generator, capable of producing expert demonstrations on-demand using motion planners. It provides rich, multi-modal sensory data, including multi-view RGB-D images, point clouds, proprioceptive states, and natural language task descriptions. A key strength of RLBench is its ability to generate countless task variations by randomizing object properties and positions, making it an ideal platform for rigorously evaluating the generalization capabilities of learned policies.

**BridgeV2**: A large-scale, real-world robot manipulation dataset designed to drive research in generalization by capturing immense visual and physical diversity. Collected across 50 unique real-world kitchen environments using a fleet of low-cost WidowX 250 6-DoF arms, the dataset was gathered through a large-scale, distributed teleoperation effort. It provides multi-view, high-resolution RGB images, proprioceptive data (joint states and gripper information), and natural language instructions for each of its over 7,200 successful task demonstrations. The defining characteristic of BridgeV2 is its unprecedented diversity; by sourcing data from a wide array of unstructured home environments, it presents policies with significant variations in lighting, object appearance, textures, and physical dynamics. This makes it an essential resource for training and evaluating general-purpose, vision-based manipulation policies that are robust to novel, unseen settings.

## A.2 DATA STATISTICS

Table 10: Comparison of Robot Manipulation Datasets

| Environment | Dataset | Number of Trajectories | Arm Type |
|---|---|---|---|
| Simulation | LIBERO | 12,000 | Single |
| | RoboTwin | 8,000 | Dual |
| | RLBench | 10,000 | Single |
| | CALVIN | 20,000 | Single |
| Real World | DROID | 76,000 | Single |
| | AGIBOT | 10,000 | Dual |
| | BridgeV2 | 14,000 | Single |

## A.3 REAL WORLD ROBOT SETTING

Our experimental platform is centered around a 7-DoF Franka Research 3 robotic arm, augmented with a UMI gripper (Chi et al., 2024) for versatile object interaction. Visual perception is provided by a RealSense D435 camera, which is statically mounted to offer a fixed, third-person perspective of the workspace. The camera captures RGB-D images at a resolution of 640×480 pixels. For intuitive data collection, we gather expert demonstrations via teleoperation using a 6-DoF 3D mouse, adapting the publicly available implementation from code [1]. The robot's control loop operates at 20 Hz. This frequency is a deliberate down-sampling from the arm's native 100 Hz controller, representing a strategic trade-off: it ensures that generated trajectories are temporally smooth and continuous, while also keeping the data volume manageable for efficient policy training. The action space is defined within SE(3), where each action is a 7-dimensional vector representing the absolute target end-effector pose: a 3D Cartesian position combined with a 4D quaternion for orientation.

To rigorously evaluate our proposed method, we have designed a comprehensive benchmark suite of six manipulation tasks. This benchmark is intentionally structured to span a wide spectrum of complexities, comprising two short-horizon tasks, two middle-horizon tasks and two long-horizon tasks. This design allows us to assess the agent's capabilities in both fundamental visuomotor control and extended, multi-stage decision-making. The short-horizon tasks primarily test for precise, reactive manipulation, while the long-horizon tasks challenge the agent's ability to plan over extended periods and robustly execute sequential sub-goals.

---

[1]https://github.com/UT-Austin-RPL/deoxys_control

For each task, we establish a standardized data collection and evaluation protocol. A dataset of 50 expert demonstrations is collected for policy learning. Subsequently, the model's performance is quantitatively assessed over 20 independent evaluation trials, each with randomized initial conditions to ensure a robust measure of generalization. The specific details and objectives of each task are outlined below:

**1) Pick up Toy:** The robot must grasp a specific target toy from the tabletop. This task tests basic object identification and manipulation.

**2) Open Drawer:** The robot is required to interact with an articulated object by approaching a closed drawer and pulling its handle to open it.

**3) Stack Cubes:** The robot needs to precisely pick up a cube and place it on top of one another.

**4) Put Toy into Basket:** This is a pick-and-place task where the robot must first pick up a specified toy and then deposit it into a nearby basket.

**5) Rank Cubes:** The robot must perceive a specific attribute of several cubes, such as size or color, and arrange them in a designated sequence.

**6) Stack Cups:** The robot's objective is to stack several cups of varying sizes in descending order, requiring it to place the largest cup first and nest the smaller ones inside it.

### A.4    TRAINING DETAILS

The Vision-Language Model (VLM) was fine-tuned for 1,000 steps with a batch size of 32 on eight 80GB NVIDIA A100 GPUs. The generalizable action expert was trained on the same hardware configuration. Its training consisted of two phases: an action pre-training phase that ran for two days with a batch size of 32,768, followed by a point cloud fine-tuning phase that ran for three days with a batch size of 256.

### A.5    SIMULATION ROBOT RESULTS

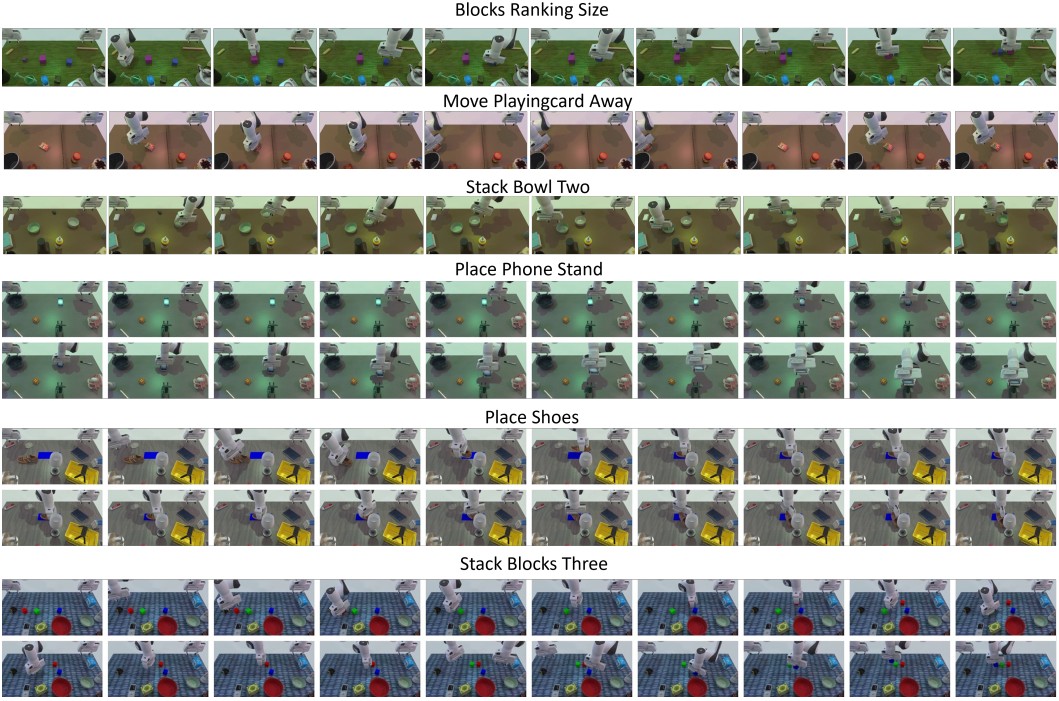

Figure 6: Simulation Robot Results

## A.6  REAL WORLD ROBOT RESULTS

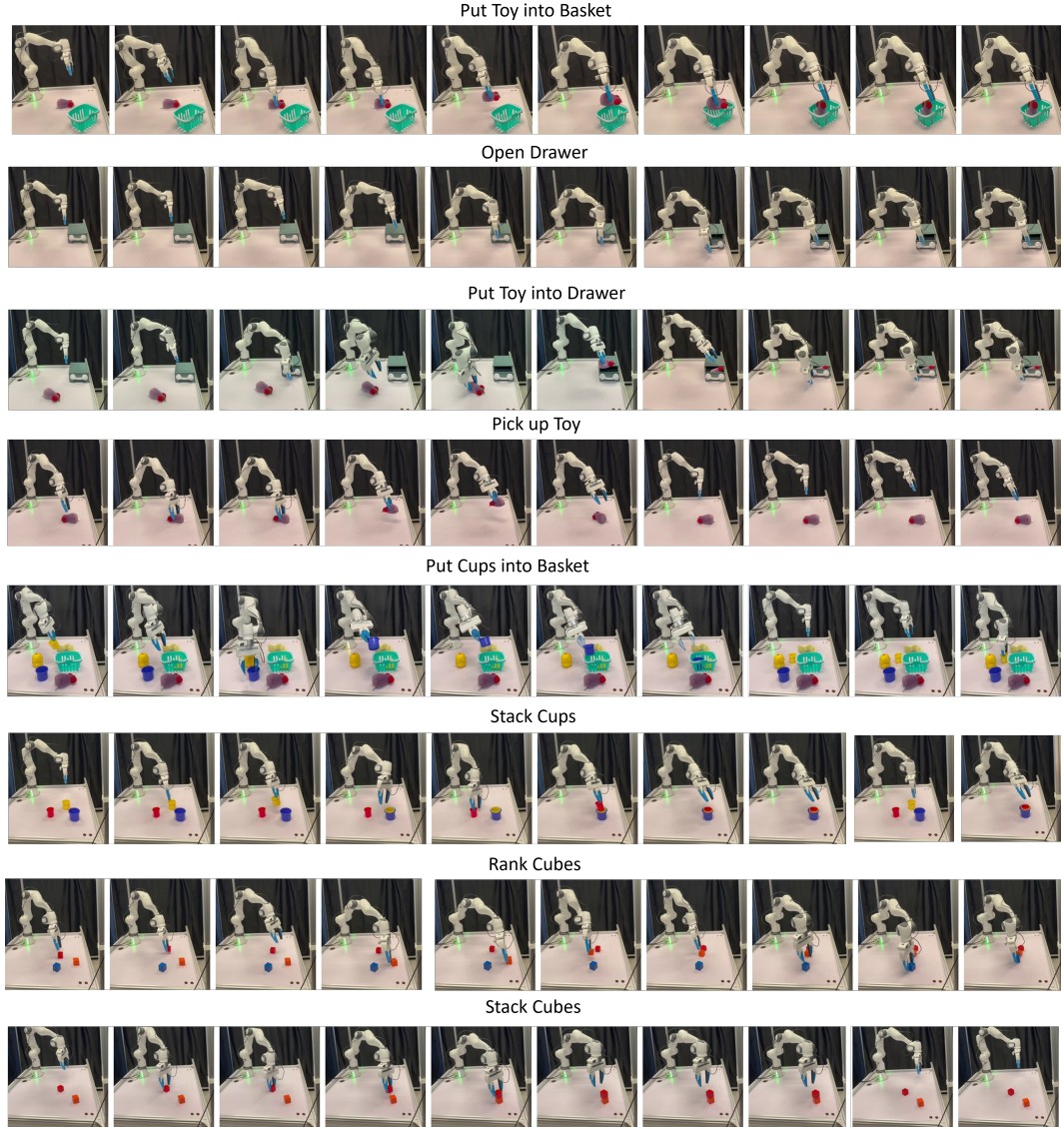

Figure 7: Real World Robot Results

