# OpenReview forum: "Bridge Thinking and Acting: Unleashing Physical Potential of VLM with Generalizable Action Expert"
_ICLR.cc/2026/Conference — ICLR 2026 Conference Withdrawn Submission_

### Official Review · Reviewer_c5bT · 2025-10-27

**Soundness:** 2
**Presentation:** 3
**Contribution:** 2
**Rating:** 4
**Confidence:** 4

**Summary:**

### Summary: Bridge Thinking and Acting

#### Research Problem
The study addresses the challenge of deploying **Vision-Language Models (VLM)** for robot control. It tackles two main issues: (1) **Poor generalization** and **catastrophic forgetting** in monolithic VLA models, and (2) **semantic ambiguity** in the interfaces of decoupled thinking/acting systems, which necessitates environment-specific fine-tuning.

#### Methodology
A new decoupled framework is proposed using **sparse 3D waypoints** as a clear, geometric interface between the VLM planner and the action executor.
* **Thinking (VLM):** Generates sparse 3D waypoints and target poses in the camera coordinate system.
* **Acting (Generalizable Action Expert):** Refines the waypoints into dense action sequences using **real-time point cloud observations**.
* **Training:** An **Action Pre-training, Pointcloud Fine-tuning (APPF)** strategy is used to first train the expert on large-scale trajectory data, then fine-tune it with point clouds for environment awareness and robustness.

#### Key Experiments
The method was validated through extensive experiments:
* **Real-World Tasks:** Evaluation on a **Franka Research 3** robot across six short, medium, and long-horizon tasks (e.g., "Stack Cubes," "Put Toy into Basket").
* **Results:** The approach demonstrated superior performance and **strong zero-shot generalization** compared to baselines.
* **Ablation:** Confirmed that the action expert **protects the VLM's language capabilities** by requiring fewer fine-tuning steps, and validated the efficiency of the APPF training paradigm.

**Strengths:**

### Strengths of the Paper

1.  **Clean Decoupling with Geometric Interface:** The use of **sparse 3D waypoints** as the interface between the VLM (thinking) and the Action Expert (acting) effectively achieves a clean, non-semantic decoupling. This is a significant improvement over existing methods that rely on ambiguous visual or semantic embeddings, making the action expert easily generalizable across different tasks and environments.

2.  **Effective Protection of VLM Knowledge:** The architecture minimizes the burden on the VLM, requiring it only to generate high-level geometric plans (waypoints) rather than low-level actions. This reduces the number of required fine-tuning steps for the VLM, effectively **preventing catastrophic forgetting** and preserving the VLM's powerful general-purpose reasoning and language understanding capabilities.

3.  **Robust and Generalizable Action Expert:** The **Action Pre-training, Pointcloud Fine-tuning (APPF)** paradigm ensures that the action expert is highly robust. By pre-training on large-scale trajectories and then refining with point cloud observations, the expert learns to perform accurate geometric refinement and demonstrates superior performance and **strong zero-shot generalization** on complex real-world manipulation tasks.

**Weaknesses:**

### Potential Weaknesses of the Paper

1.  **Reliance on Accurate 3D Information:** The core method relies heavily on the **accuracy and quality of the sparse 3D waypoints** generated by the VLM and the **real-time point cloud observations** for the Action Expert. Errors in depth estimation or camera-to-base calibration, especially in complex, unstructured environments, could lead to significant failures in trajectory refinement, limiting real-world deployment reliability.

2.  **Scalability of the VLM Fine-Tuning:** Although the method aims to *reduce* VLM fine-tuning steps, the VLM still requires **Supervised Fine-Tuning (SFT)** to learn the projection from visual observations to 3D waypoints. This SFT phase is still dependent on collecting paired visual-language-3D trajectory data, which remains a costly and non-trivial data collection problem.

3.  **Limited Handling of Dynamic Environments:** The system is primarily evaluated on structured, rigid manipulation tasks (picking, placing, stacking). The use of pre-planned trajectories, even with point cloud refinement, might be insufficient for **highly dynamic or rapidly changing scenes** where continuous, reactive replanning (beyond simple geometric correction) is necessary.

4.  **Computational Overhead of Point Cloud Processing:** The Action Expert relies on real-time point cloud observations for geometric refinement. Processing and analyzing dense point cloud data (especially for large scenes or high frame rates) adds a **significant computational cost and latency** compared to methods that operate purely on 2D images or simpler feature vectors, potentially limiting the system's reaction speed.

5.  **Simplicity of Action Primitives:** The system is structured around predicting 3D pose waypoints and gripper states. This geometric focus might inherently limit the system's ability to execute tasks that require **complex, non-geometric skills** (e.g., compliant motion, force control, precise tactile feedback) which are crucial for advanced manipulation in human-centric environments.

**Questions:**

1. **Contribution on Predict Achor Point:** This work employs the VLM model to conduct Predict Achor Point. This approach actually relies on the understanding and generalization capabilities of the pre-trained VLM model. In this section, could the author clarify where this work differs from previous works such as **ReKep** and **SOFAR**? These works also seem to perform high-level Point prediction, and similarities with these works might reduce the originality of this work in this part.

2. **VLM's understanding and generalization capabilities:** The method proposed in the article relies on the understanding and generalization capabilities of the pre-trained VLM model. However, most of the experiments presented by the author are about "grabbing individual objects" or operating on simple color blocks. **Can this experimental design truly effectively demonstrate the understanding ability of VLM and its ability to generalize to unknown objects?** Why didn't the author conduct some more complex object grasping or functional grasping tasks, or perform some tasks involving relatively rare objects (objects not present in the fine-tuning dataset) to illustrate the **necessity for VLM to exist and perform 2D point reasoning**?

3. **Overclaim:** Did the author overstate the claimed model's effectiveness and performance? In lines 82-83, the author wrote: "To the best of our knowledge, this is the first attempt to train a generalizable expert that can be deployed without requiring any task-specific fine-tuning." However, in the author's pipeline, the author explicitly mentioned conducting **Multi-Source PCD Fine-tuning** for different scenarios. Isn't this also a kind of "fine-tuning for specific task scenarios"? At the same time, the author also fine-tuned the VLM. The readers think that the author should provide explanations for these fine-tunings in their work. Are these fine-tunings based on specific scenarios and related to the author's experiments, rather than as the author emphasized, seemingly this VLA can be "used directly"?

4. **Baselines:** In the experimental part, the author compared a large number of models. However, in fact, this work of the author belongs to the dual-system series of articles within the VLA. Why didn't the author compare some works of the dual-system type, especially those that use VLM as the backbone and make some specific processing on the action head, such as **GR00T, OpenHelix, and pi0-fast**? Especially for some DP and ACT comparisons, we know that due to limitations such as parameter quantity, these works may only be used as action heads in some recent studies. Comparing these individual models with the author's work, readers may consider it inappropriate.

5. **Experiments:** Since the author's article is a dual-system project, the **execution speed**, the **frequency of updates**, and how the two systems exchange data are all topics of great concern. Could the author provide some reports on the performance in this aspect? Even a few qualitative analyses would make the article more substantial.

---

### Official Review · Reviewer_UqCe · 2025-10-30

**Soundness:** 3
**Presentation:** 2
**Contribution:** 2
**Rating:** 4
**Confidence:** 4

**Summary:**

The paper presents a hierarchical vision-language-action (VLA) model for robotics tasks. The high-level model is a VLM which does semantic reasoning - given the task description, the VLM predicts first an 2d anchoring point on the image, and then a sequence of 3d waypoints that lead to the anchor point. The low-level action expert takes in point cloud observation and a sample from the 3d waypoints, and predicts the action to achieve the waypoint. The action expert is first trained, without point cloud observation, to predict action to achieve the goal pose, called action pre-training, and then fine-tuned to produce action in consideration of the point cloud observation.

**Strengths:**

- The action expert produced by the paper shows zero-shot generalization capability. In particular, in the real-world experiments, the action expert is not trained with in-domain data, yet still has a reasonable success rate across different tasks. Generalization capability of action experts is believed to be the main bottleneck of VLA models, and the results shown in the paper seems promising.

- The idea of action pre-training and point cloud fine-tuning is novel to my knowledge. The action pre-training can quickly provide the action expert with basic prior on feasible motions, without incurring the cost for processing point clouds.

- The experiment sections come with a series of ablation studies examining different design choices made in the paper.

**Weaknesses:**

- While the action expert shows strong generalization capabilities, the VLM presented in the paper does not have the generalization capability that is on par with other existing hierarchical VLA models, e.g. [1], [2]. The VLM presented in the paper requires fine-tuning with in-domain data, whereas the VLM in [1] and [2] can generalize zero-shot to new environments and tasks. It seems like the technique proposed in the paper is not able to fully leverage the pre-trained VLM models.

- The presentation of the paper requires significant improvement. Important technical details, such as the choice of VLM model, specific network architecture for the encoders and the action expert, training hyper-parameters (optimizer, learning rate, etc.), are not included in the paper. The description of certain experimental results can be missing or confusing. This will be elaborated in the **Questions** section.

- The VLM is required to convert a 3D anchor point to a set of 3D waypoints. The transform is a function of the intrinsic parameters of the camera. This means that the VLM can only work with a specific camera, and fine-tuning is needed given a camera, e.g., with a different focal length or resolution.

- Since only 3D waypoints are communicated to the action expert, the action expert needs to figure out the orientation of the end-effector, in particular, without language description of the task. The current approach may be sufficient with the tasks presented in the paper, since they are mainly table-top pick-and-place tasks. However, the 3D waypoint interface will be limiting for more dextrous manipulation tasks in cluttered environments, which requires delicate re-orientation of the end-effector and the object being manipulated.

- Even though it is claimed, in Section 4.1, that the “in all experiments, (…) action expert is deployed in a zero-shot manner”, the action expert is trained on the RoboTwin dataset, which is the evaluation benchmark for 4.2. This is misleading and may cause uncareful readers to overestimate the generalization capabilities of the action expert.

- The action pre-training phase is designed to give the action expert a general idea of feasible motion without point cloud observation. This may be replaced by an IK solver, and the action expert is only trained to refine the IK solution based on point cloud observation. It would be nice if there is an ablation study on the necessity of action-pretraining.

- It would be nice if there is an ablation study on the use of B-splines versus straight lines.

- In Table 9, the difference between F+P+M+S and F+S does seem very significant.

- Some text descriptions are needed to explain the “results” in A.5 and A.6.


**References**

[1] Yuan, Wentao, et al. "RoboPoint: A Vision-Language Model for Spatial Affordance Prediction in Robotics." Conference on Robot Learning. PMLR, 2025.

[2] Li, Yi, et al. "HAMSTER: Hierarchical Action Models for Open-World Robot Manipulation." The Thirteenth International Conference on Learning Representations.

**Questions:**

- Which VLM model is fine-tuned to produce anchor points and 3D waypoints?

- How is the goal pose *sampled* from the end-effector trajectory? Why not deterministically choose a point on the 3D path?

- How often does the VLM generate a new end-effector trajectory? Is it per action step or per subtask? How often does the action-expert re-sample a new goal pose from the trajectory?

- When fine-tuning the VLM to predict 3D waypoints, are the waypoints given by the robot trajectories? How are the 3D waypoints subsampled from a dense trajectory?

- Is the VLM fine-tuned only on a particular in-domain dataset, or is there a “pre-training” phase where the VLM is fine-tuned to predict anchor points and waypoints for several datasets altogether?

- In Figure 2, what does L1, R1, etc. refer to?

- In Section 3.2.1, when replaying the simulation data, are the assets also included in the scene?

- What is the architecture for the encoders and diffusion model, e.g., number of layers, number of hidden units, etc.? What is the optimizer, learning rate, etc.?

- Is the point cloud input to the action expert colored or uncolored?

- What is the action space? Is the action in end-effector space or joint space? Is the output of the action expert an action chunk or single step action?

- In Section 4.2, why do \pi0 and RDT require task-specific fine-tuning after multi-task training?

- How are the numbers of training steps in Table 2 determined for the proposed approach and all baselines?

- In Table 5, how are DP3 and ACT baselines trained?

- Why compare against \pi0 and RDT for sim tasks but OpenVLA for real tasks?

- In Section 4.2.5, is the noise level referring to the noise level added during training or testing?

- How is data from the 7 dataset used during training? Are they equally weighted?

---

### Official Review · Reviewer_8Hmz · 2025-10-31

**Soundness:** 1
**Presentation:** 2
**Contribution:** 1
**Rating:** 0
**Confidence:** 5

**Summary:**

This paper proposes a two-system solution for robotics, consisting of a VLM planner and a low-level control executioner. The VLM takes in the RGB image and predict 2D points and gripper pose, which are then lifted to 3D via depth maps. The action expert takes in a 3D trajectory and the point cloud of the environment, and produces the actions.

**Strengths:**

- The problem studied (improving the capability of VLAs) is important.
- There are diverse ablation results and quantitative results (albeit on limited and simple tasks).

**Weaknesses:**

- The key contribution is not clear.
	- The paper claims that previous system dual-system suffers from "semantic ambiguities" but the claim isn't elaborated nor supported. What does "semantic ambiguities" mean?
	- The idea of using waypoints as representation for robot has been proposed before (e.g., ReKep, Hamster), the idea of 2D pointing (Gemini Robotics), and dual systems (GR00T, Hi-Robot ect).
	- Many of these works are not properly discussed and compared against. The authors discussed Hamster (Li et al, 2025), and claimed that "the semantically ambiguous nature of these intermediate representations poses a significant challenge for training the downstream action expert." In fact, this work (and many prior works like ThinkAct) use the same representation proposed in this work, i.e., 3D waypoints.
- The paper is fairly poorly written
	- Related and prior works are not faithfully discussed.
	- The paper lays down a strong claim: "this is the first attempt to train a generalizable expert that can be deployed without requiring any task-specific fine-tuning", which is not well-supported.
	- The paper devotes a lot of space, talking about "raditional Vision-Language-Action (VLA) models typically predict coordinates in the robot’s base frame...This tendency explains the poor generalization commonly seen in these models. Our approach overcomes this by predicting waypoints directly in the camera frame". This is confusing to me because as far as I know, all other methods (e.g., Hamster, Gemini Robotics) also point the 2D pixels on the image (i.e., producing the "waypoints directly in the camera frame").
     - Abstract and conclusion emphasize zero-shot deployment without further fine-tuning. Yet the real-world section states that for 6 real tasks, 50 human demos per task (300 total) were collected to fine-tune the VLM (action expert frozen). This is few shot instead of zero-shot.

- The technical method is not well-motivated or sensible.
	-  The method is quite limited and engineered to a specific problem: "First, the VLM predicts the 2D anchor point coordinates for grasping or placing a target object" (line 200). This implicitly assumes that the task is a single pick-and-place task. It's not clear how to generalize to longer horizon or more general tasks.
	- Finetuning VLM to infer 2D trajectories and target pose seem very difficult, and an out-of-distribution task for VLM, and the segmentation of the trajectory seems very arbitrary.
	- The necessities of learning the action expert is not well-supported. Why not using an off-the-shelf collision-aware motion planner such as CuRobo or using some other methods like ManipGen?
- The evaluation is fairly incomplete:
	-  The gains are moderate in some task (e.g., 0.93 vs 0.9 in Click Bell Tab 1) and underperform compared to many expert model (e..g, DP3 Tab 1). Table 4 shows Pi0 performance to be surprising low (e.g., 0.12 vs 0.86) and it's not clear whether this is actually from the method or just the fact that the authors' tabletop is more OOD.
	No comparison against similar methods like ReKep, Hamster, Hi-robot, ManipGen.
	- There is no supplemental or video results of robot execution.

**Questions:**

Please see the weaknesses.

---

### Official Review · Reviewer_yewV · 2025-11-01

**Soundness:** 2
**Presentation:** 2
**Contribution:** 3
**Rating:** 4
**Confidence:** 3

**Summary:**

This work targets robot manipulation with VLAs and proposes to go through an explicit waypoint/trajectory representation for better action predictions.
Their pipeline (1) feeds instruction text + image to a VLM, which then predicts image space waypoints.
(2) Next, using depth information, they query the VLM again with the "anchor point" to get a sparse 3D trajectory and goal pose.
(3) They densely interpolate the 3D trajectory via B-splines, which is then used by a diffusion-based action expert head to generate actions from robot state, guidance pose (sampled from trajectory), and local point cloud.

During training, they assume a pre-trained VLM/VLA and first SFT on the new waypoint task.
For the action expert head, they train in two stages: Stage 1 pretrains on pure trajectory data (huge batch size) to learn goal -> actions, and Stage 2 fine-tunes with point clouds (batch 256) for environment-aware goal + point cloud -> actions.
They train on 150k trajectories (50k sim: RoboTwin, CALVIN, LIBERO, RLBench + 100k real: DROID/AGIBOT/BridgeV2) with multi-quality point clouds (stereo, depth completion, monocular).
For evaluation, they use the RoboTwin (sim) and ManiSkill (zero-shot cross-env) benchmarks, showing performance across several tasks spanning different horizons.

**Strengths:**

Results on long horizon RoboTwin 2.0 tasks are strong (Table 1).
In short horizon tasks, they perform equal to recents works Pi-Zero and DP3.
In long horizon, they outperform the second best method Pi-Zero by nearly 4x success rate.

The idea is simple and effective, providing a nice way to unify the representation of actions across different datasets for large scale training.

The two-stage action head training is a nice little trick to speed up training.

The data and pseudo-labeling pipeline is well-engineered and could be broadly useful for VLA training.

**Weaknesses:**

Experimental section needs elaboration
- To my knowledge, the baseline models Pi-zero and DP3 don't use point cloud data while the proposed method does, making it difficult to assess whether performance gains come from the waypoint interface or simply from richer observations.
- It would be nice to clarify what modalities each method uses for Table 1 baselines

Missing comparisons to related work:
- They evaluate on the relatively new benchmark RoboTwin 2.0 and not against any of the other methods that go through intermediate representations (described in Sec 2.2).
- Their claims on lines 116-120 go without concrete evidence or ablation.

Ablations are not persuasive of the core contribution:
- The ablations don't isolate the waypoint interface contribution, nor the "action pretraining, point cloud fine-tuning" performance on the downstream task (though they do show training loss/error in Figure 5)
- The ablations/analysis do not point out why their proposed method outperforms baselines by such a large margin on long horizon tasks.

### Minor Weaknesses

- Figure 1 has typos: "Robot Sapce" and "Camera sapce" should be "Space"
- Guidance noise scale (0.1) ablation shows optimal value but doesn't analyze sensitivity or task-dependent variation
- No computational cost comparison is provided (wall-clock time, FLOPs vs baselines)

Writing needs polish
- Figure 1 is busy and it's hard to tell exactly what is new vs baseline components.
- Section 3.1 header "Lift VLM's Reasoning Power to 3D Space" is puzzling - what is "lift vlm"?

**Questions:**

- Table 1: What does the * in "Pi0*" and "RDT*" denote?
- How were the baselines trained compared to the proposed method?

---

### Author Response · Authors · 2025-11-13

We would like to express our sincere gratitude to the four reviewers for carefully reading our manuscript and providing valuable comments and suggestions. After thorough consideration, we have decided to withdraw the current version of the paper in order to further polish it. We plan to supplement additional experiments and reformulate the paper to address potential misunderstandings and improve its overall quality. Thank you again for your time and insightful feedback！

---

### Note · Authors · 2025-11-13

I have read and agree with the venue's withdrawal policy on behalf of myself and my co-authors.